# Efficacy of Life Protection Probably from Newly Isolated Bacteria against Cisplatin-Induced Lethal Toxicity

**DOI:** 10.3390/microorganisms11092246

**Published:** 2023-09-06

**Authors:** Yuka Ikeda, Naoko Suga, Satoru Matsuda

**Affiliations:** Department of Food Science and Nutrition, Nara Women’s University, Kita-Uoya Nishimachi, Nara 630-8506, Japan; tyvufkxaq1226-218@outlook.jp (Y.I.); fyb83bh720fj@gmail.com (N.S.)

**Keywords:** cisplatin, probiotics, gut microbiota, kidney, acute kidney injury, nephrotoxicity, Nrf2, SOD

## Abstract

Cisplatin may be commonly used in chemotherapy against various solid tumors. However, cisplatin has a limited safety range with serious side effects, which may be one of the dose-restraining reasons for cisplatin. A favorable therapeutic approach is immediately required for ameliorating cisplatin-induced toxicity. In the present study, the potential protective effects of certain bacteria have been investigated at the lethal dosage of cisplatin in mice experimental models. Treated under the highest dosage of cisplatin, treatment of certain commensal bacteria could significantly increase the survival rate. In addition, our findings revealed that probiotic supplementation of these bacteria could result in the attenuation of the damage appearance on the kidney as well as the alteration of several antioxidant-related gene expressions, including *SOD1*, *SOD2*, *SOD3*, *Nrf2*, and/or *HO-1* genes in the high dosage of cisplatin-treated mice. In short, acute kidney injury in mice was induced by a single dose of cisplatin 11 or 15 mg/kg intraperitoneally. Then, peroral administration of newly isolated bacteria could protect against the cisplatin-induced injury, probably by decreasing oxidative stress. Therefore, the data shown here might suggest that the usage of certain probiotic supplementation could contribute to the life protection of patients suffering from severe toxicity of cisplatin. However, the molecular mechanisms need to be further explored.

## 1. Introduction

Cisplatin has been widely used in the treatment of solid tumors [1]. However, about 30% of cisplatin-administered patients have suffered from severe kidney dysfunction, especially termed acute kidney injury [1,2]. The acute alterations in kidney function may often lead to death [3,4]. Therefore, cisplatin has a relatively narrow safety range and, unfortunately, also shows serious dose-limiting other side effects, including neurotoxicity, ototoxicity, nausea, and vomiting, in addition to serious nephrotoxicity [5]. Hence, it is urgent to explore novel therapeutic procedures to protect patients, reduce acute kidney injury, and/or maintain the quality of life (QOL) in cisplatin-treated patients. Although the pathophysiological origin of cisplatin nephrotoxicity has been deeply studied, the molecular mechanism of acute kidney injury has not been clarified yet. Evidence suggests that oxidative stress, inflammation, necrosis, pyroptosis, and/or apoptosis may play crucial roles in cisplatin-induced nephrotoxicity [1]. In particular, reactive oxygen species (ROS) might be involved in the pathogenesis of nephrotoxicity [6]. There are few strategies or tactics for competently preventing cisplatin-induced acute kidney injury [7]; therefore, this has still been the main limiting factor restricting cisplatin’s clinical application.

Accumulating evidence has suggested the imperative role of gut microbiota in the pathogenesis of inflammatory diseases, including acute kidney injury. Gut microbiota may subsequently become a promising therapeutic target for chemotherapy-induced nephrotoxicity, including cisplatin-induced acute kidney injury, which could mediate the protective effects of certain bacteria as beneficial probiotics against cisplatin-induced acute kidney injury. Gut microbiota might also provide a scientific basis for future clinical applications of the specific probiotics to treat cisplatin-induced acute kidney injury. Actually, chemotherapy is frequently accompanied by an altered composition of gut microbiota known as dysbiosis [8], which may be defined by the loss of worthy bacteria, such as bifidobacilli and/or lactobacilli species, and/or by replacing the pathogenic microorganisms. Therefore, cisplatin-induced nephrotoxicity may also be associated with a change in the gut microbiota, which may follow the damage of gut barrier function and/or an increase in uremic matters in bodies [9]. Actually, it has been revealed that certain probiotics could work as a good mediator to restore the intestinal microbiota composition in cisplatin-induced nephrotoxicity [10]. In addition, modification of the gut microbiota through probiotics could lessen the progression of kidney disease as well as the renal damage of cisplatin-induced nephrotoxicity, which might be considered a beneficial intervention to stimulate butyrate production, remove uremic toxins and increase renal function [10]. The precise molecular and precise biological mechanisms of probiotics with health-advantageous properties regrettably remain unidentified. Possibly, certain probiotics could alter the gut microbiota and suppress the growth of pathogenic microorganisms. Furthermore, it has been proposed that alteration of the intestinal flora through probiotics could slow down the progression of various kidney diseases [11,12]. However, again, the underlying mechanisms have not been well elucidated yet [13]. 

Initially, the aim of this study was to seek out the special bacteria possessing the protective effect against cisplatin-induced acute kidney injury. For this purpose, looking for specific bacteria that could suppress oxidative stresses and beneficially change gut microbiota has been a struggle. Recently, we have fortunately isolated several clones of bacteria with high antioxidant potential from various fruits and/or vegetable materials (personal communication, unpublished data, and manuscript in preparation). Here, we have evaluated the effects of two clones on these bacteria. After some preliminary experiments, we hypothesized that supplementation of either Bact A or Bact B could prevent or reduce the damage by cisplatin. In conclusion, after all sections, we have confirmed that both bacteria could protect the lives of individuals suffering from cisplatin-induced severe toxicity.

## 2. Materials and Methods

### 2.1. Mice

We purchased 6-week-old male ICR mice from Charles River Laboratories Japan, Inc. (Kanagawa, Japan) and kept them in conditions of 24 °C, humidity 55%, and a 12-h dark/light cycle. All mice could access food and water ad libitum. These animal experiments were directed in accordance with the “Guidelines for Animal Experiments at Nara Women’s University” and the “Standards for the Care and Storage of Laboratory Animals and the Alleviation of Pain (Approval Number 21-01)”.

Exp. 1: After acclimatization, the mice were divided into three groups: Cis (cisplatin-induced kidney injury, n = 8), Cis/Bact A (cisplatin-induced kidney injury/bacteria A, n = 5), and Cis/Bact B (cisplatin-induced kidney injury/bacteria B, n = 5). The Cis group mice drank 1% fructose water, the Cis/Bact A group mice drank 1% fructose water with bacteria A, and the Cis/Bact B group mice drank 1% fructose water with bacteria B. On day 1, all mice received 11 mg/kg B.W. of cisplatin intraperitoneally. We measured the mice’s life span for 2 weeks. 

Exp. 2: After acclimatization, the mice were divided into groups: the Ct (control, n = 4), Cis (cisplatin-induced kidney injury, n = 6), Cis/Bact A (cisplatin-induced kidney injury/bacteria A, n = 6), Cis/Bact B (cisplatin-induced kidney injury/bacteria B, n = 6). The Ct and Cis group mice drank 1% fructose water, the Cis/Bact A group mice drank 1% fructose water with bacteria A, and the Cis/Bact B group mice drank 1% fructose water with bacteria B. On day 5, all mice received 15 mg/kg B.W. of cisplatin intraperitoneally, and the Ct group received the same amount of saline intraperitoneally. On day 9, all mice were sacrificed, and we collected samples of blood, kidney, liver, and spleen. 

### 2.2. Materials

Cisplatin and fructose were purchased from Fujifilm Wako Pure Chemicals Co. (Tokyo, Japan). The cisplatin (CAS No. 033-20091) was dissolved in saline solution and made 10 mg cisplatin/10 mL saline for the injection. And before and after the injection, we had an 8-h non-watering period.

### 2.3. Real-Time PCR

The total RNA from the kidney was extracted following the RNAiso Plus manual (Takara Bio Inc., Shiga, Japan), making 0.5 µg/µL RNA samples. The reverse transcription reaction was performed following the ReverTra Ace qPCR RT Master Mix manual (TOYOBO Co., Ltd., Tokyo, Japan) to obtain cDNA samples. 

The PCR reaction was performed using the LightCycler Nano instrument (Roche Diagnostics K.K., Tokyo, Japan). The PCR reaction solution was made following the SYBR Green real-time PCR method. We mixed THUNDERBIRD SYBR qPCR Mix (TOYOBO Co., Ltd., Tokyo, Japan), sterile water, 0.1 mM forward primer, 0.1 mM reverse primer, and cDNA solution. The sequences of primers are shown in Table 1.

### 2.4. Western Blotting

The protein of the kidney was extracted by homogenization with RIPA buffer. We obtained supernatant by centrifuging the homogenate at 15,000× g/min for 15 min (Tabletop micro-cooled centrifuge Model3500, Kubota, Osaka, Japan) twice and mixed it with sample buffer to adjust to about 1 mg/mL protein concentration. 

To separate proteins, we used SDS-PAGE and transferred them to nylon membranes (Immobilon-P, Merck KGaA, Darmstadt, Germany). The membranes were blocked with 3% skim milk in TBST solution overnight at 4 °C. The next day, it reacted with primary antibodies Nrf2 (Nuclear factor erythroid 2-related factor 2, CUSABIO, Houston, TX, USA) or HO-1 (Heme oxygenase 1, Gene Tex, Irvine, CA, USA) at 1 h and peroxidase-conjugated goat anti-rabbit secondary antibodies (Cell Signaling, Danvers, MA, USA) at 1 h. After that, the membranes reacted with luminescence (ImmunoStar, FUJIFILM Wako Pure Chemicals Co.) and were detected by the ImageQuant LAS500 (G.E. Healthcare Japan Com., Tokyo, Japan). Each band was quantified by using ImageJ, and the relative ratio of protein expression was analyzed by GAPDH (Glyceraldehyde 3-phosphate dehydrogenase, FUJIFILM Wako Pure Chemicals Co.) as an internal control protein.

### 2.5. Statistical Analysis

All data are shown as means  ±  standard error (S.E.). Data analysis of this study was performed using GraphPad Prism version 5.0 (GraphPad Software, Inc., San Diego, CA, USA) by two-way ANOVA Bonferroni or one-way ANOVA Dunnett tests and Pearson’s correlation analysis. A *p*-value  <  0.05 is considered statistically significant. 

## 3. Results 

### 3.1. Survival Curve of High-Dose Cisplatin-Treated Mice with or without Newly Isolated Bacteria

To test the toxic effect of cisplatin, a high dose of cisplatin (cisplatin 20 mg/kg body weight) was adopted. We tested the effects of cisplatin using a single administration of the cisplatin dose. The maximum tolerated doses of cisplatin in rodents have been reported to range from 6 to 12 mg/kg of body weight, depending on the mouse strain [14]. It has also been reported that ten percent lethal dose values may range from 8 mg/kg to 15 mg/kg for rodents [15]. In the present experiment, two types of bacteria were used for mice as probiotics. As shown in the survival curve in Figure 1, half of the mice died within 2 weeks of the treatment of high-dose cisplatin without probiotics (Figure 1). However, oral Bact A or Bact B supplementation as probiotics could completely improve the survival rate, even under such a lethal dose of cisplatin (Figure 1). Moreover, cisplatin-treated mice with the Bact supplementation appeared very active compared to the mice without Bact supplementation (Video movie available on demand). However, the results of the survival rate were not significant by the calculation of Kaplan–Meier analysis (*p* > 0.05), probably because of the low number of mice used. 

### 3.2. Characteristics of the High-Dose Cisplatin-Treated Mice

The experimental design of the further analytical experiment is shown in Figure 2. As body weight loss may be a main characteristic of cisplatin-treated mice, we first detailed the changes in body weights and/or food intake to examine the benefit of the probiotics (Figure 3A,B). After a few days post cisplatin injection, the body weight was significantly decreased in all the cisplatin-treated groups (Figure 3A). Comparable to the body weight loss, the food intake was decreased in these groups (Figure 3B). In the present experimental condition, however, there was no significant difference in the alterations in body weight and/or food intake among the Cis, Cis/Bact A, and Cis/Bact B groups (Figure 3A,B). No mice died during the entire experiment time (within 4 days after cisplatin treatment) with this cisplatin dose (cisplatin 15 mg/kg body weight). The relative kidney weight was increased in the Cis group than that of the Ct group (Figure 3C). There was no difference in the relative kidney weight among the Cis, Cis/Bact A and Cis/Bact B groups (Figure 3C). Likewise, the relative spleen weight was not significantly different among the Cis, Cis/Bact A and Cis/Bact B groups, which were all significantly lower compared to that of the Ct group (Figure 3D). Macroscopic findings of the kidney are shown (Figure 4). There was almost no difference in the appearance among all groups in the present experiment (Figure 4).

### 3.3. Analyses for Genes and Protein Expression in the High-Dose Cisplatin-Treated Mice

Kidney injury molecule-1 (KIM-1) is one of the biomarkers of kidney dysfunction [16]. In the present experiment, the *Kim-1* gene expression in the Cis group mice was significantly higher than that in the Ct group mice. There was no difference in the *Kim-1* gene expression among the Cis, Cis/Bact A and Cis/Bact B groups (Figure 5A). Inversely, all the *superoxide dismutase 1 (Sod1)*, *superoxide dismutase 2 (Sod2)*, and *superoxide dismutase 3 (Sod3)* gene expressions in the Cis group mice were significantly lower than those in the Ct group mice (Figure 5B–D). The *Sod1* and *Sod2* gene expressions of the Cis/Bact A and Cis/Bact B group mice were slightly increased compared to that of the Cis group mice but remained still lower than those of the Ct group mice (Figure 5B,C). As for *Sod3* gene expression, there was almost no difference among the Cis, Cis/Bact A and Cis/Bact B groups (Figure 5D).

In this experiment, Western blotting showed that the protein expression of nuclear factor erythroid 2-related factor 2 (Nrf2) was noticeably upregulated in the Cis/Bact A and/or Cis/Bact B group mice (Figure 6A,B), suggesting that Bact A and/or Bact B might have a potential for life-protection by attenuating oxidative stresses and/or inflammatory injury via the regulation of the Nrf2 signaling pathway [17]. However, the protein expression of heme oxygenase 1 (Ho-1) was not so much upregulated, either in the Cis/Bact A or in the Cis/Bact B group mice (Figure 6A,C).

## 4. Discussion

The cisplatin doses used for the chemotherapy experiment against tumors in a range of 5 to 10 mg/kg might be acceptable in mice [18]. A cisplatin dose of 10 mg/kg may exhibit severe toxicity to the present mice with a 30% mortality (Ikeda personal communication). Cisplatin-treated mice have been reported to gradually lose body weight after cisplatin injection [19] and would be dead in one or two weeks. Consistently, we found this severe toxicity in mice after the treatment with high-dose (11, 15 mg/kg) cisplatin (Figure 1 and Figure 2A,B). In this high dose of cisplatin treatment, Bact A and Bact B could exhibit some protection to the life of mice (Figure 1). Probiotic treatment prior to the high-dose-bolus cisplatin challenges was also mildly protective, similar to the present result (Ikeda personal communication). 

In the present study, we observed considerable life-protective effects from Bact A or Bact B on cisplatin-induced severe toxicity. Although there was no difference in the body weight between the Cis group and Cis/Bact A and B groups, it is probable that substantial protection might have occurred in the kidney. It is also plausible that both Bact A and Bact B could disturb the growth of such harmful bacteria in the gut. For example, it has been reported that chemotherapy-associated gastrointestinal toxicity might be associated with an increased abundance of LPS-producing bacteria [20,21]. In addition, probiotics therapy could exert anti-inflammatory activity in dextran sulfate sodium (DSS)-induced colitis by the modulation of the phosphatidylinositol-3 kinase (PI3K)/AKT pathway [22]. Since *Akkermansia muciniphila* acts on the PI3K/AKT signaling pathway [23], it is possible that Bact A or B might control the quantity of *Akkermansia muciniphila*. 

Bact A, Bact B, and some fecal microbiomes of the mice treated with these bacteria have been identified by 16S rRNA gene sequencing, which may not be published here due to patents pending. However, the possible beneficial effects of these bacteria might be related to the anti-inflammatory and/or antioxidative effects [24]. Correspondingly, it has been shown that probiotics with *Lactobacillus* strains could have anti-inflammatory and/or antioxidative effects [25]. In addition, the *Bifidobacterium* species might also have anti-inflammatory and/or antioxidative effects against inflammatory bowel diseases [26]. Likewise, *Clostridium butyricum* could also amend inflammation in the colon by decreasing the levels of inflammatory cytokines [27]. In these ways, probiotics and/or prebiotics appear to be considered effective protectants for several host organs, which might also improve cisplatin-induced severe toxicity. However, both Bact A and Bact B have been identified as neither *Lactobacillus*/*Bifidobacterium* species nor *Clostridium butyricum* strains. Both the Bact A and Bact B employed in the present experiments are derived from bananas, and they are different species (unpublished data). The molecular mechanisms by which Bact A or Bact B could employ the beneficial effects against cisplatin-induced severe toxicity are now under investigation and should be further explored in the future.

It has been reported that cisplatin-induced AKI model mice may exhibit hypertrophy of the kidney. [28]. Consistently, the relative weight of the kidney/body was also increased in the Cis group mice (Figure 3C). There was almost no difference among the Cis, Cis/Bact A, and Cis/Bact B groups, suggesting that Bact A or Bact B could not block the hypertrophy (Figure 3C and Figure 4). It has been reported that cisplatin-induced AKI model mice cause atrophy of the spleen. Consistently, the relative weight of the spleen/body was also decreased in the Cis group mice, which presented little difference among the Cis, Cis/Tre, and Cis/H-Tre groups (Figure 3D). 

To reveal the life protection mechanism of Bact A or Bact B, we focused on the roles of intracellular antioxidant pathways. It has been reported that many antioxidants could demonstrate efficacy in reducing the severity of AKI and delay the progression of several kidney diseases [29]. In line with this, the result of Western blotting shows that the protein expression of Nrf2 and/or Ho-1, which are eminent molecules capable of being involved in the antioxidants pathway, were more or less high in the Cis/Bact A and Cis/Bact B group mice rather than that in the Cis group mice (Figure 6A–C). Moreover, the *Sod1* and *Sod2* gene expression—well-known antioxidant enzymes that are related to the reduction of reactive oxygen species (ROS)—had changed in the Cis/Bact A and/or Cis/Bact B group mice compared to that in the Cis group mice (Figure 5B,C). It has been shown that these molecules may have protective effects against oxidative stresses via the autophagy mechanism [30]. However, probable antioxidant therapies using these bacteria should be more explored. 

Some limitations to this study should be noted. The consequences of the analyses in a small sample size study should be deciphered with attention because the lesser sample size of experiments might often lead to non-significant outcomes. In particular, we cannot exclude the possibility that the consequences of this study overstated the correctness due to the small number of mice used in all subgroups. In addition, as the consequences of the present study have been achieved by using a mouse model, the application of our conclusion to humans should be carefully deliberated. Moreover, it should be examined whether cisplatin treatment with probiotics using Bact A or Bact B could affect the efficacy of the chemotherapy. Forthcoming investigations with a larger amount of animals to direct the above concerns would become more informative.

## 5. Conclusions

Several data in the present study have indicated to elucidate the potential mechanisms of certain probiotics in the prevention and/or inhibition of cisplatin-induced severe toxicity. This concept might also be useful for the development of treatment against various inflammatory kidney diseases. Systematically, the protective effects might be in part related to the control of antioxidant mechanisms, which might also be associated with anti-inflammatory effects.

## Figures and Tables

**Figure 1 microorganisms-11-02246-f001:**
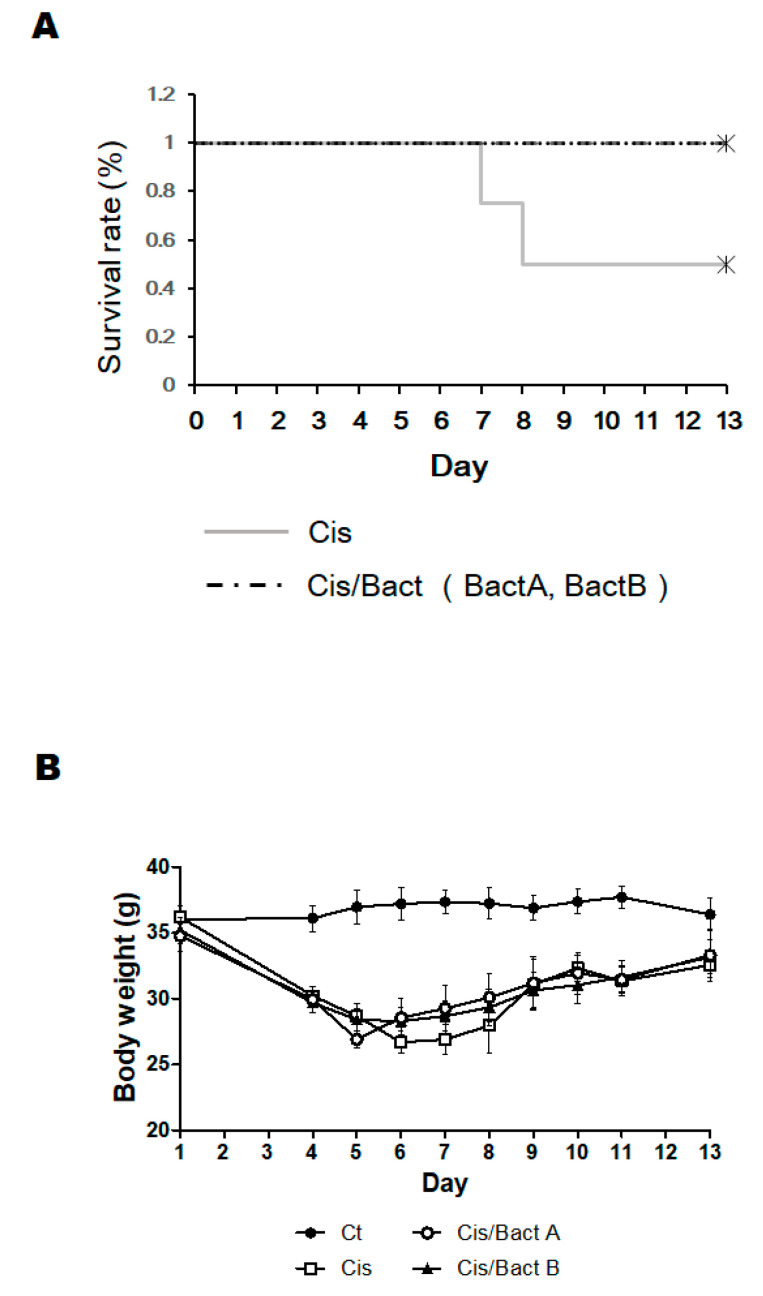
The effects of Bacteria on prolonging mice lives. (**A**) Survival data of mice for 13 days using the Kaplan–Meier method. The gray line shows the Cis group, and the black dotted line shows Cis/Bact A and Cis/Bact B. Both the Cis group mice and Cis/Bact group mice received 11 mg/kg B.W. of cisplatin intraperitoneally. (**B**) Body weights were measured almost every day during the experiment. Ct group (black circle), Cis group (white square), Cis/Bact A group (white circle), and Cis/Bact B group (black triangle). Values are expressed as the mean ± S.E. n = 3–8/group.

**Figure 2 microorganisms-11-02246-f002:**
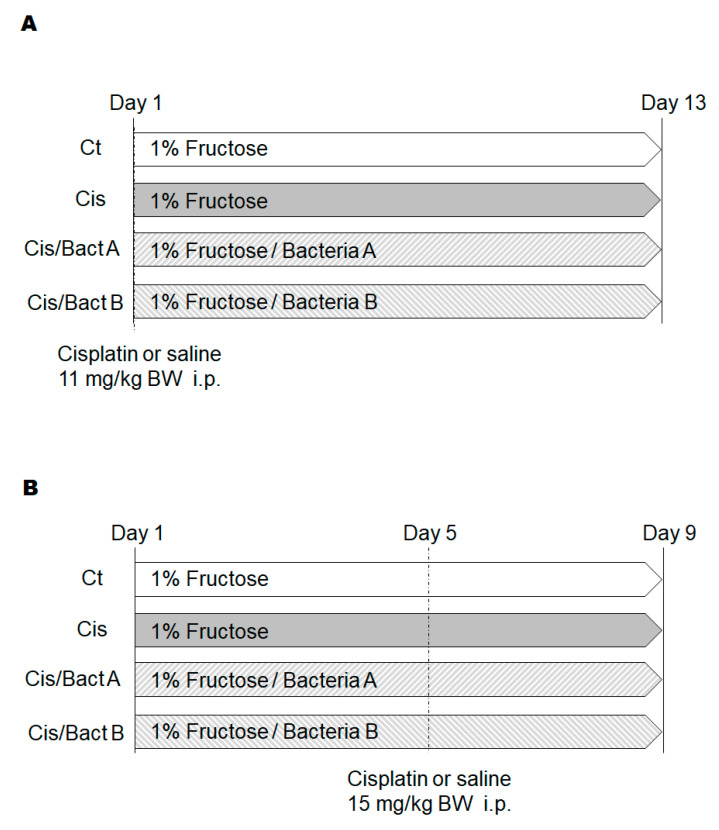
Study design. (**A**) Male ICR mice (6 weeks old) were separated into four groups: Ct (control, n = 4), Cis (cisplatin-induced kidney injury, n = 6), Cis/Bact A (cisplatin-induced kidney injury/bacteria A, n = 6), Cis/Bact B (cisplatin-induced kidney injury/bacteria B, n = 6). On day 5, the Ct group received 15 mg/kg B.W. of saline intraperitoneally, and the Cis, Cis/Bact A and Cis/Bact B groups received 15 mg/kg B.W. of cisplatin intraperitoneally. All mice were sacrificed on day 9. (**B**) Male ICR mice (6 weeks old) were divided into four groups: Ct (control, n = 4), Cis (cisplatin-induced kidney injury, n = 6), Cis/Bact A (cisplatin-induced kidney injury/bacteria A, n = 6), Cis/Bact B (cisplatin-induced kidney injury/bacteria B, n = 6). On day 5, the Ct group received 15 mg/kg B.W. of saline intraperitoneally, and the Cis, Cis/Bact A and Cis/Bact B groups received 15 mg/kg B.W. of cisplatin intraperitoneally. All mice were sacrificed on day 9.

**Figure 3 microorganisms-11-02246-f003:**
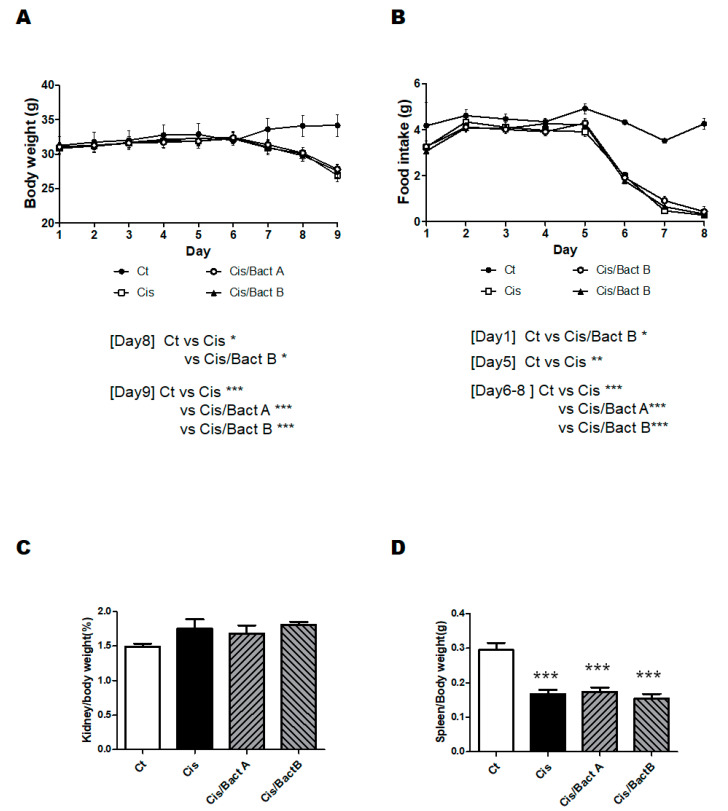
Body weight and some tissue weights. (**A**) Body weights were quantified every day during the experiment. Ct group (black circle), Cis group (white square), Cis/Bact A group (white circle), and Cis/Bact B group (black triangle). Values are stated as the mean ± S.E. n = 4–6/group. The data were verified by two-way ANOVA. (* *p* < 0.05, *** *p* < 0.005, vs. Ct group); (**B**) The food intake was quantified every day during the experiment. Ct group (black circle), Cis group (white square), Cis/Bact A group (white circle), and Cis/Bact B group (black triangle). Values are stated as the mean ± S.E. n = 4–6/group. The data were tested by two-way ANOVA. (* *p* < 0.05, ** *p* < 0.01, *** *p* < 0.005, vs. Ct group); (**C**) kidney/body weights were quantified when mice were dissected. Ct group (white), Cis group (black), Cis/Bact A group (right upper diagonal), and Cis/Bact B group (left upper diagonal). Values are stated as the mean ± S.E. n = 4–6/group. The data were verified by one-way ANOVA; (**D**) spleen/body weights were quantified when mice were dissected. Ct group (white), Cis group (black), Cis/Bact A group (right upper diagonal), and Cis/Bact B group (left upper diagonal). Values are stated as the mean ± S.E. n = 4–6/group. The data were verified by one-way ANOVA (*** *p* < 0.005, vs. Ct group).

**Figure 4 microorganisms-11-02246-f004:**
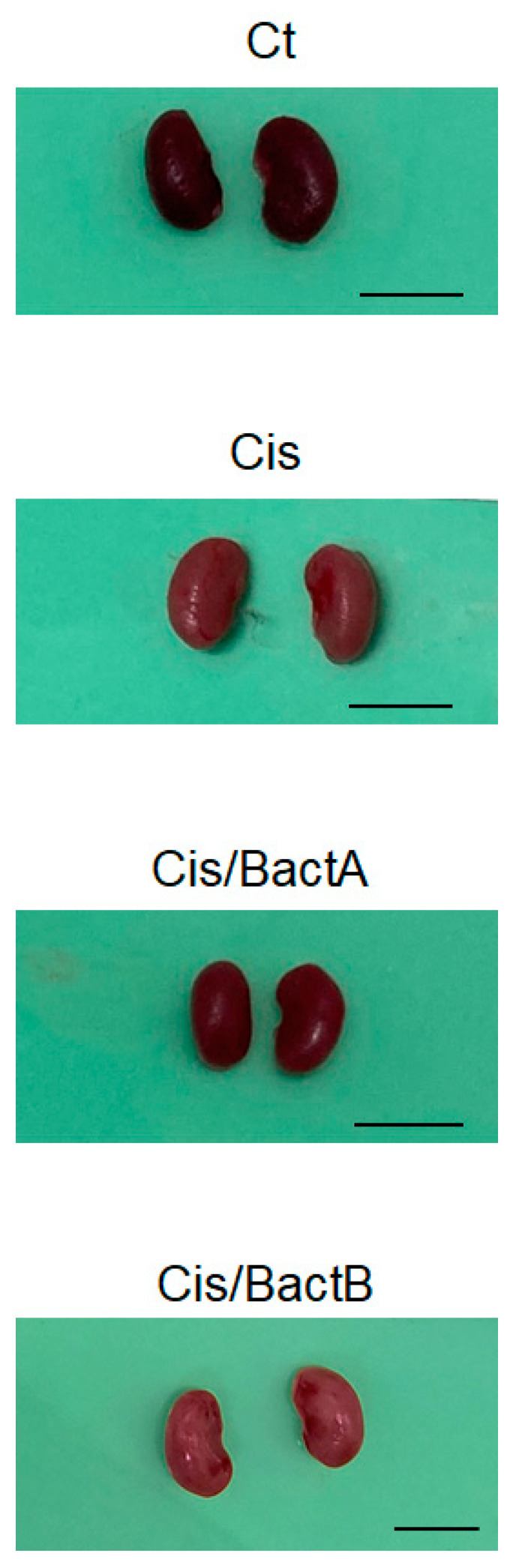
The color of the kidney. The kidney photos of each group represent mice. Each scale bar shows 10 mm. As shown in images of Cis or Cis/BactB, the color of the kidneys became pale or whitened due to severe inflammation.

**Figure 5 microorganisms-11-02246-f005:**
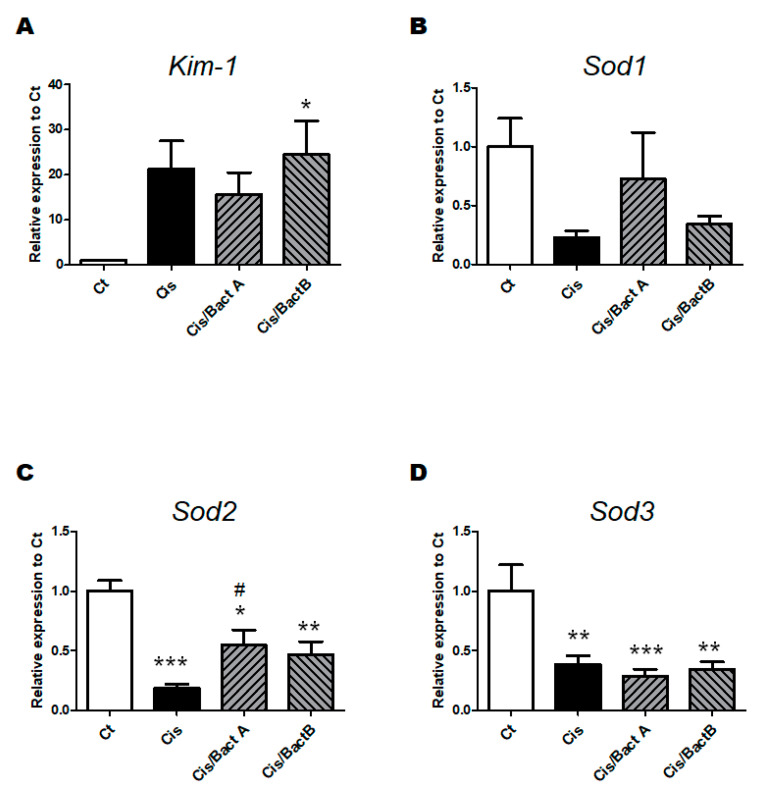
The gene expression of the kidney. (**A**) The mRNA expression of Kim-1 was quantified and normalized to that of beta-actin by RT-PCR. Ct group (white), Cis group (black), Cis/Bact A group (right upper diagonal), and Cis/Bact B group (left upper diagonal). Values are stated as the mean ± S.E. n = 4–6/group. The data were certified by one-way ANOVA (* *p* < 0.05, vs. Ct group). Kim-1: Kidney injury molecule-1. (**B**) The mRNA expression of Sod1 was quantified and normalized to that of beta-actin by RT-PCR. Ct group (white), Cis group (black), Cis/Bact A group (right upper diagonal), and Cis/Bact B group (left upper diagonal). Values are stated as the mean ± S.E. n = 4–6/group. The data were verified by one-way ANOVA. Sod1: Superoxide dismutase 1. (**C**) The mRNA expression of Sod2 was also quantified and normalized to that of beta-actin by RT-PCR. Ct group (white), Cis group (black), Cis/Bact A group (right upper diagonal), and Cis/Bact B group (left upper diagonal). Values are conveyed as the mean ± S.E. n = 4–6/group. The data were verified by one-way ANOVA (* *p* < 0.05, ** *p* < 0.01, *** *p* < 0.005, vs. Ct group, ^#^
*p* < 0.05, vs. Cis group). Sod2: Superoxide dismutase 2. (**D**) The mRNA expression of Sod3 was quantified and normalized to B-actin by RT-PCR. Ct group (white), Cis group (black), Cis/Bact A group (right upper diagonal), and Cis/Bact B group (left upper diagonal). Values are expressed as the mean ± S.E. n = 4–6/group. The data were verified by one-way ANOVA (** *p* < 0.01, *** *p* < 0.005, vs. Ct group). Sod3: Superoxide dismutase 3.

**Figure 6 microorganisms-11-02246-f006:**
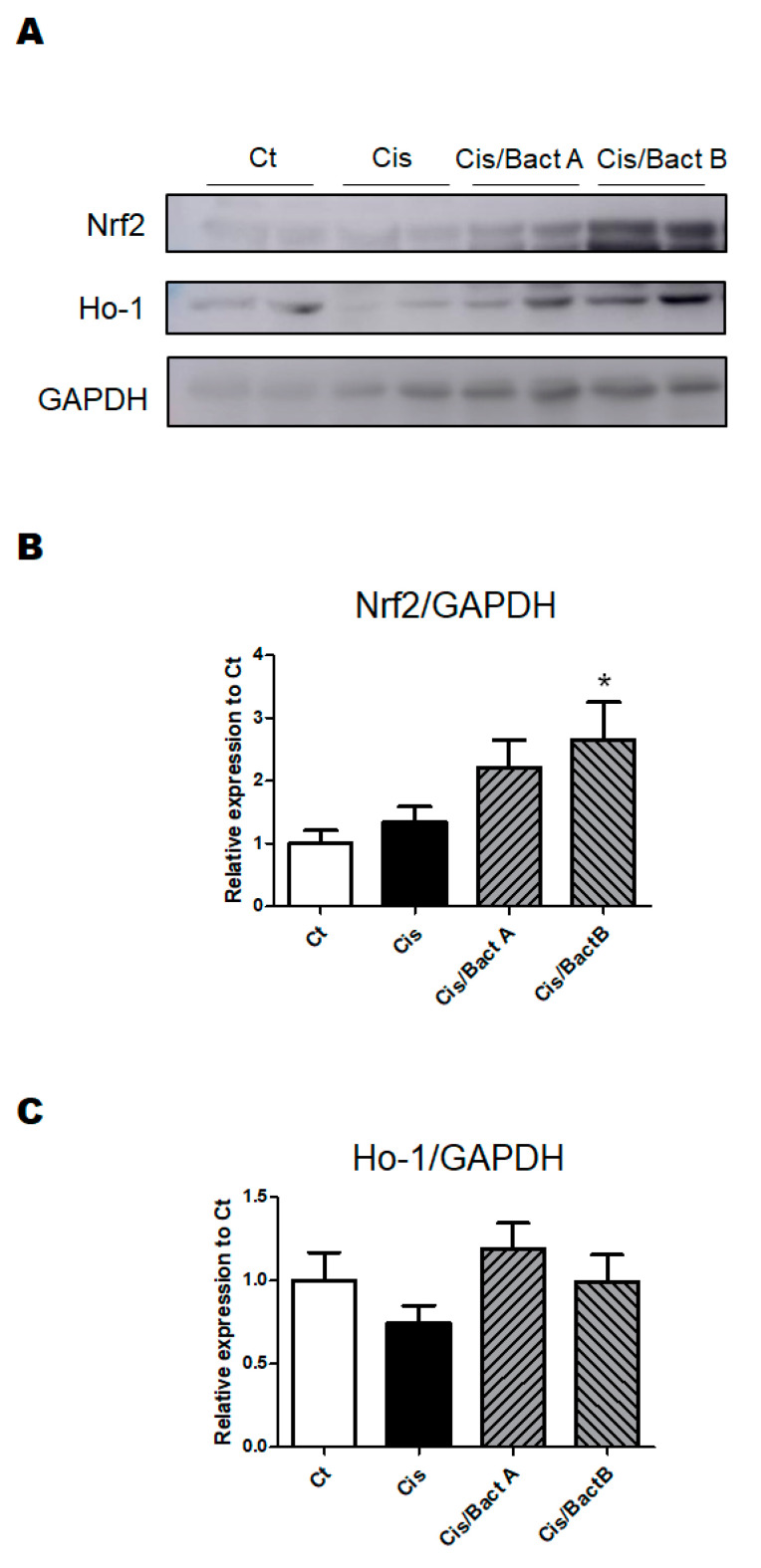
The protein expression of the kidney. (**A**) The image of Nrf2, HO-1 and GAPDH protein expressions. (**B**) The protein expression of Nrf2 was calculated and normalized to GAPDH by Westen blot. Ct group (white), Cis group (black), Cis/Bact A group (right upper diagonal), and Cis/Bact B group (left upper diagonal). Values are expressed as the mean ± S.E. n = 4–6/group. The data were verified by one-way ANOVA (* *p* < 0.05, vs. Ct group). Nrf2: Nuclear factor erythroid 2-related factor 2. (**C**) The protein expression of HO-1 was quantified and normalized to GAPDH by Westen blot. Ct group (white), Cis group (black), Cis/Bact A group (right upper diagonal), and Cis/Bact B group (left upper diagonal). Values are stated as the mean ± S.E. n = 4–6/group. The data were verified by one-way ANOVA (* *p* < 0.05, vs. Ct group). HO-1: Heme oxygenase 1.

**Table 1 microorganisms-11-02246-t001:** Primer sequences for real-time PCR.

Gene	Forward Primer (5′→3′)	Reverse Primer (5′→3′)
β-actin	TTCTACAATGAGCTGCGTGTG	CTTTTCACGGTTGGCCTTAG
Kim-1	ACATATCGTGGAATCACAACGAC	ACAAGCAGAAGATGGGCATTG
Sod1	AAGAGAGGCATGTTGGAGACC	CGGCCAATGATGGAATGCTC
Sod2	TGGAGAACCCAAAGGAGAGTTG	CAGGCAGCAATCTGTAAGCG
Sod3	CTGACAGGTGCAGAGAACCTC	GCGTGTCGCCTATCTTCTCA

## Data Availability

Not applicable.

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
