# Peer review of "Efficacy of Life Protection Probably from Newly Isolated Bacteria against Cisplatin-Induced Lethal Toxicity"

_microorganisms, 2023, doi:10.3390/microorganisms11092246_

Round 1

Reviewer 1 Report

Cisplatin is commonly used in chemotherapy for solid tumors but has limited safety and serious side effects, which can restrict its dosage Yuka Ikeda et al investigated the protective effects of specific bacteria in mice treated with a lethal dose of cisplatin. Certain commensal bacteria significantly increased survival rates and attenuated kidney damage. Furthermore, probiotic supplementation altered antioxidant gene expressions, suggesting a potential life-protection strategy for patients experiencing severe cisplatin toxicity. Although this manuscript was interesting, more details are needed to improve the quality for publication.

Line 92. Any specific reasons for choosing male mice?

Line 98-112. Did the author repeat the Exp.1 and Exp.2?

Line 114. Catalog number should be provided.

Line 151. The type of bacteria used in the present study should be stated in the methods and results section.

Line 171. The histology of kidney data is important to show the effect of cisplatin on mice. And the authors measured the liver enzymes that metabolize this type of drug?

Figure 6A, the band of GADPH is a bit wired. Did the authors use different concentrations of proteins as it looks like a huge difference?

Author Response

Reviewer1

Cisplatin is commonly used in chemotherapy for solid tumors but has limited safety and serious side effects, which can restrict its dosage Yuka Ikeda et al investigated the protective effects of specific bacteria in mice treated with a lethal dose of cisplatin. Certain commensal bacteria significantly increased survival rates and attenuated kidney damage. Furthermore, probiotic supplementation altered antioxidant gene expressions, suggesting a potential life-protection strategy for patients experiencing severe cisplatin toxicity. Although this manuscript was interesting, more details are needed to improve the quality for publication.

Line 92. Any specific reasons for choosing male mice?

Because, it has been reported that estrogen receptor has a potential of cellular protection in the experiment of cisplatin-induced toxicity.

Effect of daidzein on cisplatin-induced hematotoxicity and hepatotoxicity in experimental rats. Karale S, Kamath JV.Indian J Pharmacol. 2017 Jan-Feb;49(1):49-54. doi: 10.4103/0253-7613.201022.

Estradiol protects hair cells from cisplatin-induced ototoxicity via Nrf2 activation. Adachi M, Yanagizono K, Okano Y, Koizumi H, Uemaetomari I, Tabuchi K.Redox Rep. 2023 Dec;28(1):2161224. doi: 10.1080/13510002.2022.2161224.

Line 98-112. Did the author repeat the Exp.1 and Exp.2?

One reason was to get the reproducibility of this experimental result.

In addition, we had considered that the successful result was fortunately derived from the relatively lower dose of cisplatin in Exp.1. Therefore, a more increased dose of cisplatin had been employed in Exp.2.

Line 114. Catalog number should be provided.

Certainly. The catalog number of cisplatin has been provided in the text.

Line 151. The type of bacteria used in the present study should be stated in the methods and results section.

We have further mentioned about the information of the bacteria used in the discussion section.

Line 171. The histology of kidney data is important to show the effect of cisplatin on mice. And the authors measured the liver enzymes that metabolize this type of drug?

The histology of kidney data was confusing. Hence, we omitted the data this time. Liver function could frequently influence the renal function. Just in cases, we measured the liver enzymes.

Figure 6A, the band of GADPH is a bit wired. Did the authors use different concentrations of proteins as it looks like a huge difference?

We have replaced the image of GAPDH in Figure 6A with another one.

Reviewer 2 Report

Cisplatin
 is a highly effective chemotherapeutic agent that has been used for more than 50 years for a variety of cancers; however, its use is limited by toxicity, including nephrotoxicity. Cisplatin accumulates in the 
kidney tubules and causes AKI through various mechanisms, including DNA damage, oxidative stress, and apoptosis

Detailed comments:

1. The lack of description of the bacterial strains or reference to the description of these bacteria in the literature. Such anonymity of the bacteria used, prevents publication of this manuscript.

2. At what concentration were the probiotics administered (log cfu g-1)? Or was it a freeze-dried powder, then give the dose (mg/kg B.W./day)? But you write that you dissolved the bacteria in water and fructose. Did each mouse drink the same amount of water with the probiotic?

3.Why were the mice kept alive for only 9-14 days. After all, the practice is about 42 days and probiotics are given up to week 6. Too short a period is one of the reasons for the rejection of this manuscript.

4 A histopathological examination should have been performed for this study. The absence of this examination is one of the reasons for rejection of the manuscript.

Author Response

Reviewer2

Cisplatin is a highly effective chemotherapeutic agent that has been used for more than 50 years for a variety of cancers; however, its use is limited by toxicity, including nephrotoxicity. Cisplatin accumulates in the kidney tubules and causes AKI through various mechanisms, including DNA damage, oxidative stress, and apoptosis. 

Detailed comments:

  1. The lack of description of the bacterial strains or reference to the description of these bacteria in the literature. Such anonymity of the bacteria used, prevents publication of this manuscript.

As mentioned in the discussion section, Bact A and Bact B are not lactic acid bacteria sp. All the studies such as Gram stain, microscopic morphology, and colony appearance of Bacteria have been included in the manuscript submitted elsewhere (unpublished data). Both Bact A and Bact B derived from banana, which has been additionally described in the discussion section.

  1. At what concentration were the probiotics administered (log cfu g-1)? Or was it a freeze-dried powder, then give the dose (mg/kg B.W./day)? But you write that you dissolved the bacteria in water and fructose. Did each mouse drink the same amount of water with the probiotic?

Prior to the experiments, we found that the growth of both bacteria at RT would make a platou level in these conditon of DSS/k-carrageenam or fructose drinking water. Hence, both bacterial concentrations for the peroral administration might be approximately 1 × 106 CFU ml-1 during the usage period. We had found that each mouse drank almost similarly at the same amount of them.

  1. Why were the mice kept alive for only 9-14 days. After all, the practice is about 42 days and probiotics are given up to week 6. Too short a period is one of the reasons for the rejection of this manuscript.

Maybe, you are right in the view point for direct clinical application of this experiment. Although this would be preliminary experiment, the important hint for the clinical application might be definitely suggested from the result of this experiment even with too short period of observation. We think that is the real reason why this manuscript should be published. In addition, we thought that a higher dose of cisplatin may break the protection of these bacteria. We had confirmed that mice injected higher dose of cisplatin might die earlier than mice with lower dose of cisplatin. After dead of mice, we could not perform several protein- and/or blood- analyses. Therefore, the surgery of Exp.2 was performed at the day earlier than that of Exp.1, for getting cisplatin-damage-comparison between mice with bacteria and without bacteria at the same days after cisplatin treatment.

  1. A histopathological examination should have been performed for this study. The absence of this examination is one of the reasons for rejection of the manuscript.

Absolutely, this is correct. However, the histology of kidney data was confusing among and/or within groups. Hence, we omitted the data this time. At present, we think it probably from our technical error.

Reviewer 3 Report

In their article “Life Protective Effect from Certain Probiotics against Cisplatin-Induced Lethal Toxicity,” Ikeda et al. report that probiotic supplementation attenuated the adverse effects of high-dose cisplatin treatment in mice. The study is interesting because cisplatin is commonly given to tumor patients, and the authors demonstrate that probiotic supplements could protect those suffering from severe toxicity of cisplatin. The findings of this study are novel and very interesting. Though the paper is well written, there are some concerns. This paper can be accepted for publication after minor revision.

Comments

1. Abstract; include the methodological protocol.

2. In the introduction, line 85 - Explain the rationale for using two clones selectively. 3.  In the experiments, mice drank 1% fructose water with Bact A or B. Explain the cell concentration of Bact A or B probiotic strains dissolved in the water. How do the authors determine the efficacy of the probiotics used?

4. In the results, Figure 1, the written part about the dosage used is very confusing. Rewrite the section on what dose of cisplatin is used in this study. Then may include available literature on the dosage of cisplatin.

5. If the authors inject 11 mg/kg cisplatin, how much cisplatin accumulates in the body of mice?

6.  Exp 2 – rational for choosing a different dose of 15 mg/kg of cisplatin and the surgery on day 9, explain.

7. Figure 3D: Discrepancy between figure data and legend Spleen/body weights.

 The authors did a fine job in the overall writing of the manuscript however some sections can be improved, see comments, and authors should be checked for grammatical corrections.

Author Response

Reviewer3

In their article “Life Protective Effect from Certain Probiotics against Cisplatin-Induced Lethal Toxicity,” Ikeda et al. report that probiotic supplementation attenuated the adverse effects of high-dose cisplatin treatment in mice. The study is interesting because cisplatin is commonly given to tumor patients, and the authors demonstrate that probiotic supplements could protect those suffering from severe toxicity of cisplatin. The findings of this study are novel and very interesting. Though the paper is well written, there are some concerns. This paper can be accepted for publication after minor revision.

Thank you so much for the good evaluation to our manuscript.

Comments

  1. Abstract; include the methodological protocol.

Accordingly, some methodological protocol has been included in the abstract. Thank you so much.

  1. In the introduction, line 85 - Explain the rationale for using two clones selectively.

We thought the protection efficacy of those bacteria might be different. Although Bact A and Bact B were not similarly lactic acid sp, both Bact A and Bact B were derived from banana. However, the 16S rRNA gene sequencing revealed that they are different species (unpublished data), which have been mentioned in the discussion section.

  1. In the experiments, mice drank 1% fructose water with Bact A or B. Explain the cell concentration of Bact A or B probiotic strains dissolved in the water. How do the authors determine the efficacy of the probiotics used?

As a result, the efficacy was ultimately evaluated by the death or alive comparison. Prior to the experiments, we found that the growth of both bacteria at RT would make a platou level in these conditon of DSS/k-carrageenam or fructose drinking water. Hence, both bacterial concentrations for the peroral administration might be approximately 1 × 106 CFU ml-1 during the usage period. We had found that each mouse drank almost similarly at the same amount of them.

  1. In the results, Figure 1, the written part about the dosage used is very confusing. Rewrite the section on what dose of cisplatin is used in this study. Then may include available literature on the dosage of cisplatin.

Sorry, it seems confusing. One reason for two experiments with different doses of cisplatin was to get the reproducibility of these experimental result. In addition, we had considered that the successful result from exp.1 was fortunately and/or occasionally derived from the relatively lower dose of cisplatin in Exp.1. Therefore, a more increased dose of cisplatin had been employed in Exp.2, for evaluating the more protection capability of bacteria.

  1. If the authors inject 11 mg/kg cisplatin, how much cisplatin accumulates in the body of mice?

Sorry, we did not know the accumulation of cisplatin in the body of mice. Therefore, we injected a higher dose of 15mg/kg cisplatin in the Exp.2 for evaluating the protection capability of bacteria.

  1. Exp 2 – rational for choosing a different doses of 15 mg/kg of cisplatin and the surgery on day 9, explain.

As mentioned above, we thought that a higher dose of cisplatin may break the protection of these bacteria. We had confirmed that mice injected higher dose of cisplatin might die earlier than mice with lower dose of cisplatin. After dead of mice, we could not perform several protein- and/or blood- analyses. Therefore, the surgery of Exp.2 was performed at the day earlier than that of Exp.1.

  1. Figure 3D: Discrepancy between figure data and legend Spleen/body weights.

We have improved the Figure 3D. Therefore, the discrepancy has been also amended.

  The authors did a fine job in the overall writing of the manuscript however some sections can be improved, see comments, and authors should be checked for grammatical corrections.

According to these suggestions, we have gone over the text/abstract and amended typos and grammatical errors as much as possible to improve the manuscript more helpful to the readers.

Reviewer 4 Report

REVIEW

 Dear authors,

Please consider the following comments to improve the content of your manuscript before publication. 

MAJOR CONCERN:

1.      The Title must be modified, since the type of probiotic microorganism used is not specified, if they are recently isolated or commercial strains, specify.

2.      In the 1. Introduction, more information is missing about the possible mechanism by which probiotics can help improve the health of patients treated with chemotherapy, focused on improving kidney function.

3.      Section 2.1 Mice is deficient in information about the administration of bacteria A and B in the 2 experimental models used. What was the amount of CFU administered for each bacterium? For how many days was the probiotic treatment administered in Experiment 1? Why did you use water with 1% Fructose instead of just water? How did you ensure that the mice ingested the desired amount of CFU?

4.      I think they should add a Figure with the experimental scheme of Experiment 1.

5.      What was the justification for using 2 different doses of cisplatin for Experiments 1 and 2?

6.      The number of animals used in the survival test by the Kaplan-Meier method is very small, at least 8 animals should have been used for all groups.

7.      They mention that they extracted blood, liver and spleen samples, but at no point in the methodology do they mention what they were used for. Later in the results a graph appears with the relative weight of the spleen, but nothing is mentioned about the blood and liver samples.

8.      In section 2.2 Materials, they do not describe the characteristics of the 2 microorganisms used in this work (Bact A and Bact B), if they are indicating the word "probiotics" in the Title of the article, then they must have a reference where they are mentioned. their characteristics or describe them in this section (Gram, LAB's, origin of isolation).

9.      Figure 1 must be accompanied by the monitoring of the weight of the mice, since the mice treated with Bact A and Bact B there is no difference in the % of survival (100%), it implies that the 2 strains are the same for behavior. Why didn't they extend the survival curve more days? It lacks information on the X and Y axes.

10.   Figure 2 should be in section 2. Materials and Methods.

11.   In Figure 3B, the abbreviation Cis/Tre and Cis/H-Tre are not mentioned.

12.   In Figure 3D, the word Spleen/ is missing on the Y axis.

13.   The images in Figure 4 are not relevant, since the quantitative information is represented in Figure 3C. Delete Figure 4 or leave it as supplementary material.

14.   The results of Figures 5 and 6 seem that there is no beneficial effect when the treatment with Bact A and Bact B is administered in the murine model of cisplatin, with the blood obtained I recommend that you evaluate inflammatory cytokines and with the liver do an evaluation hisopathological for greater evidence of its protective effect on health as mentioned.

MINOR CONCERN:

1.      The Abbreviations section must be before the References section.

2.      Write in italics the scientific names of the microorganisms in the References section.

 Please amend the requested comments and submit the revision file.

Author Response

Reviewer4

Dear authors,

Please consider the following comments to improve the content of your manuscript before publication. 

Thank you so much for the valuable comments.

MAJOR CONCERN:

  1. The Titlemust be modified, since the type of probiotic microorganism used is not specified, if they are recently isolated or commercial strains, specify.

The title has been improved accordingly. In short, the previous title “Life protective effect from certain probiotics against cisplatin induced lethal toxicity” has been replaced with the new title “Efficacy of life protection from newly isolated bacteria against cisplatin induced lethal toxicity”.

  1. In the1. Introduction, more information is missing about the possible mechanism by which probiotics can help improve the health of patients treated with chemotherapy, focused on improving kidney function.

That is a very good point. The introduction section has been improved accordingly.

  1. Section 2.1 Miceis deficient in information about the administration of bacteria A and B in the 2 experimental models used. What was the amount of CFU administered for each bacterium? For how many days was the probiotic treatment administered in Experiment 1? Why did you use water with 1% Fructose instead of just water? How did you ensure that the mice ingested the desired amount of CFU?

Prior to the experiments, we found that the growth of both bacteria at RT would make a platou in these conditon of DSS/ k-carrageenam drinking water. Hence, both bacterial concentrations for the peroral administration might be approximately 1 × 106 CFU ml-1 during the usage period.

  1. I think they should add a Figure with the experimental scheme of Experiment 1.

The experimental scheme of Experiment 1 has been also included in Figure 2.

  1. What was the justification for using 2 different doses of cisplatin for Experiments 1 and 2?

The doses of cisplatin used for experiments were approximately determined from the literature of similar experiment. We thought that a higher dose of cisplatin may break the protection of these bacteria. In addition, we had confirmed that mice injected higher dose of cisplatin might die earlier than mice with lower dose of cisplatin. After dead of mice, we could not perform several protein- and/or blood- analyses. Therefore, the surgery of Exp.2 was performed at the day earlier than that of Exp.1.

  1. The number of animals used in the survival test by the Kaplan-Meier method is very small, at least 8 animals should have been used for all groups.

Statistically, your suggestion is correct. With this small number of animals used in the experiment, we could not insist on the “significant” efficacy, but the relevant efficacy. Considering the importance of this result, further reproducibility of Kaplan-Meier method with considerable number of animals and huge cost/sacrifice might be compulsory for the clinical application, however, which could be authorized by the present result.

  1. They mention that they extracted blood, liver and spleen samples, but at no point in the methodology do they mention what they were used for. Later in the results a graph appears with the relative weight of the spleen, but nothing is mentioned about the blood and liver samples.

We had prepared their samples for the subsequent additional analyses depending from the result of protein/gene analyses, and even though required by the manuscript reviewers. The relative weights of the spleen have been shown, as they were somewhat different between groups.

  1. In section 2.2 Materials, they do not describe the characteristics of the 2 microorganisms used in this work (Bact A and Bact B), if they are indicating the word "probiotics" in the Title of the article, then they must have a reference where they are mentioned. their characteristics or describe them in this section (Gram, LAB's, origin of isolation).

  1. Figure 1must be accompanied by the monitoring of the weight of the mice, since the mice treated with Bact A and Bact B there is no difference in the % of survival (100%), it implies that the 2 strains are the same for behavior. Why didn't they extend the survival curve more days? It lacks information on the X and Y axes.

Certainly. We have amended the Figure 1, accordingly.

  1. Figure 2 should be in section 2. Materials and Methods.

All the analytical experiments (Exp.1 and Exp.2) had been designed after evaluating the Kaplan-Meier method.  Therefore, we think this is the right position of Figure 2.

  1. In Figure 3B, the abbreviation Cis/Treand Cis/H-Tre are not mentioned.

The figure 3B has been improved. Thank you so much.

  1. In Figure 3D, the word Spleen/is missing on the Y axis.

The figure 3D has been also improved accordingly. Thank you so much.

  1. The images in Figure 4are not relevant, since the quantitative information is represented in Figure 3C. Delete Figure 4 or leave it as supplementary material.

We think that these images might show the visual kidney damage as another evidence of AKI. In addition, we have added the explanation in the figure legend.

  1. The results of Figures 5 and 6 seem that there is no beneficial effect when the treatment with Bact A and Bact B is administered in the murine model of cisplatin, with the blood obtained I recommend that you evaluate inflammatory cytokines and with the liver do an evaluation hisopathological for greater evidence of its protective effect on health as mentioned.

Exactly, that is the point. We could not have elucidated the precise mechanism of life-protection from Bact A or Bact B, currently. As you pointed out, we are now trying to clarify the protection mechanism with Bact A and Bact B by analyzing the cytokines, antioxidants, and/or immune-histopathological studies. However, there was a mild but significant beneficial effect on the SOD2 expression with Bact A, as shown in Figure 5C.

MINOR CONCERN:

  1. The Abbreviations section must be before the Referencessection.

Probably, it might be arranged by editorial process. Thank you.

  1. Write in italics the scientific names of the microorganisms in the Referencessection.

We have improved them as much as possible, accordingly. Thank you.

 Please amend the requested comments and submit the revision file.

Again, thank you so much for the valued comments.

Round 2

Reviewer 1 Report

The authors have addressed the majority of my concerns, and the revised version of the manuscript has improved greatly. There are still a few concerns that have to be addressed before considering publication.

1.     The scale bar has to be stated in the figure’s legend.

2.     The level of GADPH in Figure 6a varies among different groups, how does the author determine the different expression of those proteins?

Author Response

Reviewer1

The authors have addressed the majority of my concerns, and the revised version of the manuscript has improved greatly.

Thank you so much.

There are still a few concerns that have to be addressed before considering publication.

  1. The scale bar has to be stated in the figure’s legend.

Accordingly, the scale bar has been stated in the legend of Figure4.

  1. The level of GADPH in Figure 6a varies among different groups, how does the author determine the different expression of those proteins?

In Figure6, the amount of loading samples might vary. Therefore, the expression of protein has been assessed as a ratio to the GAPDH level in each sample. In addition, we had employed the expression of beta-actin level as another loading control, which had suggested the similar result to that with this GAPDH level.

Reviewer 2 Report

Dear Authors,

After reading your explanations and making corrections - I have no further comments. 

Author Response

Reviewer2

After reading your explanations and making corrections - I have no further comments. 

Thank you.

Reviewer 4 Report

REVIEW

Dear authors,

The work was corrected and most of the observations made were addressed, providing enough information so that the writing has greater quality and relevance, however, it is necessary to attend to the following observations before its publication:

-        In section 2.2 Materials, they do not describe the characteristics of the 2 microorganisms used in this work (Bact A and Bact B), if they are indicating the word "probiotics" in the Title of the article, then they must have a reference where they are mentioned. their characteristics or describe them in this section (Gram, LAB's, origin of isolation).

-        Line 157: modify the title of section 3.1 Survival curve of high dose cisplatin-treated mice with or without certain probiotics, now it should agree with the title of the work “newly isolated bacteria”.

Please amend the requested comments and submit the revision file.

Author Response

Reviewer4

The work was corrected and most of the observations made were addressed, providing enough information so that the writing has greater quality and relevance, however, it is necessary to attend to the following observations before its publication:

Thank you so much for the good evaluation.

-        In section 2.2 Materials, they do not describe the characteristics of the 2 microorganisms used in this work (Bact A and Bact B), if they are indicating the word "probiotics" in the Title of the article, then they must have a reference where they are mentioned. their characteristics or describe them in this section (Gram, LAB's, origin of isolation).

This suggestion makes sense. The information of the microorganisms used in this work has been stated as much as possible in the text of introduction or discussion. In addition, the title has been also improved without using the word “probiotics”.

-        Line 157: modify the title of section 3.1 Survival curve of high dose cisplatin-treated mice with or without certain probiotics, now it should agree with the title of the work “newly isolated bacteria”.

The title of section 3.1 has been altered with using the word “newly isolated bacteria”, accordingly. Thank you so much.